# When Less is More: Vaping Low-Nicotine vs. High-Nicotine E-Liquid is Compensated by Increased Wattage and Higher Liquid Consumption

**DOI:** 10.3390/ijerph16050723

**Published:** 2019-02-28

**Authors:** Jorien Smets, Frank Baeyens, Martin Chaumont, Karolien Adriaens, Dinska Van Gucht

**Affiliations:** 1Thomas More University of Applied Sciences, Molenstraat 8, 2018 Antwerp, Belgium; jorien.smets@thomasmore.be; 2Faculty of Psychology and Educational Sciences, KU Leuven—University of Leuven, Tiensestraat 102, 3000 Leuven, Belgium; frank.baeyens@kuleuven.be (F.B.); karolien.adriaens@kuleuven.be (K.A.); 3Department of Cardiology, Erasme University Hospital, Université Libre de Bruxelles, 1050 Brussels, Belgium; martin.chaumont@ulb.ac.be

**Keywords:** electronic cigarette, trends in vaping, nicotine

## Abstract

(1) Background: Previous research (Van Gucht, Adriaens, and Baeyens, 2017) showed that almost all (99%) of the 203 surveyed customers of a Dutch online vape shop had a history of smoking before they had started using an e-cigarette. Almost all were daily vapers who used on average 20 mL e-liquid per week, with an average nicotine concentration of 10 mg/mL. In the current study, we wanted to investigate certain evolutions with regard to technical aspects of vaping behaviour, such as wattage, the volume of e-liquid used and nicotine concentration. In recent years, much more powerful devices have become widely available, e-liquids with very low nicotine concentrations have become the rule rather than the exception in the market supply, and the legislation has been adjusted, including a restriction on maximum nicotine concentrations to 20 mg/mL. (2) Methods: Customers (*n* = 150) from the same Dutch online vape shop were contacted (to allow a historical comparison), as well as 274 visitors from the Facebook group “Belgian Vape Bond” to compare between groups from two different geographies and/or vaping cultures. (3) Results: Most results were in line with earlier findings: Almost all surveyed vapers were (ex-)smokers, had started (80%) vaping to quit smoking and reported similar positive effects of having switched from smoking to vaping (e.g., improved health). A striking observation, however, was that whereas customers of the Dutch online vape shop used e-liquids with a similar nicotine concentration as that observed previously, the Belgian vapers used e-liquids with a significantly lower nicotine concentration but consumed much more of it. The resulting intake of the total quantity of nicotine did not differ between groups. (4) Conclusions: Among vapers, different vaping typologies may exist, depending on subcultural and/or geographic parameters. As a consequence of choosing low nicotine concentrations and consuming more e-liquid, the Belgian vapers may have a greater potential to expose themselves to larger quantities of harmful or potentially harmful constituents (HPHCs) released during vaping.

## 1. Introduction

In previous research [1], our goal was to assess the profile of a typical online vape shop customer. Customers of a representative medium-size online retailer in the Netherlands (www.e-cig4u.nl) were, after their online purchase, referred to an online questionnaire (time window December 2015–January 2016). Results were largely in line with data gathered through surveys of brick-and-mortar vape shop customers [2,3] and data from convenience sampling of vapers who visited e-cigarette discussion fora or smoking cessation websites [4,5,6,7]. To summarize, we found that almost all online vape shop customers (99%, *n* = 201 out of 203) had a history of smoking, having started smoking at around 15 years of age and having continued to do so for, on average, 30 years. Most of them had made at least one attempt to quit (on average around five), while trying different smoking cessation aids, but had reported those (with the exception of the e-cigarette) to be non-effective. Almost everyone who participated was a regular vaper (17% were dual users) having used the e-cigarette for more than two years. Those vapers indicated that they were/are using e-cigarettes mainly to quit smoking/remain smoking abstinent because vaping is healthier than smoking and because of financial reasons. They used open system second-generation or variable wattage third-generation devices and consumed on average 19.50 mL e-liquid per week with an average nicotine concentration of almost 10 mg/mL. A substantial majority reported that their health had improved (83%) since they had started vaping and that they were able to quit smoking (81%) or substantially reduce their cigarette consumption (85%). More than half believed that vaping is not harmful to not harmful at all, but a substantial minority still had vaping-related health concerns. 

Data from a subsequent pilot study [8], questioning respondents of an online vape forum (time window October 2017), showed a pattern that was not observed in studies until then: Respondents indicated they were vaping at high wattage (89% > 20 Watt, of which 33% > 60 Watt) using coils with low resistance (92% < 1 Ohm, of which 65% < 0.5 Ohm) with low-nicotine e-liquid (99% < 12 mg/mL, of which 74% < 4 mg/mL), but consuming large volumes of those e-liquids (87% > 4 mL per day, of which 40% > 10 mL per day). An important question then arises: Is this a pattern of a certain subculture/regional phenomenon or rather a more general historical evolution, not restricted to specific subcultures/specific regions? If indeed vaping at high-wattage while consuming large volumes of e-liquid with low-nicotine concentrations is (becoming) a new trend some new questions arise. For example, do these vapers use lower nicotine concentrations as an attempt to reduce their nicotine intake? Could it be the case that they are not aware that due to self-titration [9,10,11], they will probably largely fail to lower their daily nicotine intake because they may compensate the low nicotine concentrations by consuming (much) more liquid (see data pilot study [8])? And finally, do they realize that as a direct consequence of consuming larger quantities of e-liquid, and hence taking in larger quantities of harmful or potentially harmful constituents (HPHCs) released during vaping, they probably also absorb larger quantities of HPCPs, e.g., [12]?

In this new survey, we wanted to investigate certain evolutions with regard to technical aspects of vaping behaviour, such as wattage, the volume of e-liquid used and nicotine concentration. To compare between vapers from two different geographies and/or vaping cultures, we have included two groups: First, a Dutch sample comparable to the one we investigated two years earlier (to look at historical evolutions between 2015–2016 and 2018), and secondly, a Belgian sample of dedicated vapers participating in a Facebook forum (to look at regional/subcultural differences).

## 2. Materials and Methods 

### 2.1. Participants

For data collection, we first chose the same representative medium-size dedicated online retailer in the Netherlands as used in one of our previous studies (www.e-cig4u.nl) [1], which is an “average”, not high-end, but specialized middle-of-the-road online vape shop with an extensive and diverse product offer. Customers who made an online purchase in this vape shop between 8 January and 28 February 2018 were invited to participate (see Figure 1). Within this time window, the online retailer had 1514 customers. Two hundred and thirty of them (around 15%) started filling out our questionnaire, of whom 150 (around 65%) completed it. A second pool of participants was contacted through an announcement in Dutch and in French (posted on 29 January 2018) on the Facebook group of Union Belge Pour La Vape/Belgische Damp Bond (Belgian Vape Bond, http://www.ubv-bdb.be/). This non-profit organization (3624 members on 24 December 2018) promotes and defends vaping, including taking legal action. In total, 356 persons started filling out our questionnaire, with 274 completing it (around 77%). 

The first sample thus consisted of 150 customers of the Dutch online vape shop and was named the “Dutch Current” group (the group in our previous sample [1] will be called the “Dutch Historical” group). The group of 274 Belgian vapers contacted via Facebook belonging to the second sample was named, in analogy, the “Belgian Current” group.

### 2.2. Measures

An online questionnaire, based on the one used in an earlier study [1], was made in Qualtrics [13]. The first part of this questionnaire consisted of questions assessing background information including gender, age, nationality, education, profession, and monthly income. 

The second part contained questions about smoking (history). Participants were asked whether they had smoked (and if so, more than 100 tobacco cigarettes in their life), at what age they had started and if they were still smoking at the moment of questioning. Current smokers were asked how long they had been smoking, how many cigarettes they smoked per day, if they had undertaken any quit attempts and, if so, how many. Former smokers were asked how many quit attempts they had needed before they had quit successfully. 

Both current smokers, with previous quit attempts, and former smokers then indicated which smoking cessation aids they had used and which had been successful. The options were nicotine patches/gum/tablets, inhaler, mouth spray, smoking cessation medication, professional help from, for example, a tobacco counsellor/psychologist/general practitioner (GP), e-cig, or none/will power. They indicated how long their most successful quit period had lasted, whether they currently used smoking cessation aids, and how long they had smoked in total. Finally, current smokers rated their motivation to quit, and a list of possible complaints they experienced due to smoking (such as throat aches, tendency to cough, bad taste, bad smell, etc.) on a scale from 1 (never), over 3 (sometimes), to 5 (always), and filled out the Fagerström Test for Cigarette Dependence (FTCD) [14].

The third part of the questionnaire contained questions about vaping, starting with “Did you ever use an e-cig?” If yes, participants were asked to indicate their current vaping frequency by choosing one of three predefined categories: every day, a few days a week, not anymore. Current vapers were asked how long they had been using e-cigs. Then, all participants who had ever used an e-cig were asked to indicate the reasons for starting using e-cigs (*to quit smoking, to smoke less, because it is healthier than tobacco cigarettes, dual use, financial reasons, out of curiosity, to pass time, because smoking is prohibited in certain places, different flavours, others do it, others offered it*). Current vapers were asked to indicate why they still used e-cigs (same predefined categories plus *because it is tasty and I enjoy it*). 

Next, current vapers indicated how they usually inhaled (*mouth-to-lung inhalation, direct lung inhalation or both*), how often they experienced a dry hit and if so, what they did to avoid it. We asked them which nicotine concentration they currently used and whether they had adapted the nicotine concentration since they had started vaping (*same, reduced or increased*). We also asked how much e-liquid they currently consumed in an average week and whether the amount of e-liquid had changed since they had started vaping (*same, reduced or increased*). Then, they were asked how many times per day they used their e-cig, how many puffs they took per vaping bout and per day, which mixture of propylene glycol (PG) and vegetable glycerin (VG) they usually vaped, which coil resistance they usually used, and at what wattage (or voltage) they usually vaped. Subsequently, they indicated whether they had changed the wattage since they had started vaping (*same, reduced or increased*), and if so, why they had reduced it (e.g., *I use an e-liquid with more nicotine, I want less vapour, I want a colder vapour*) or increased it (e.g., *I use an e-liquid with less nicotine, I want more vapour, I want a warmer vapour*). 

Current vapers were then asked what type of device they used, a device without the possibility to regulate wattage/voltage/temperature, a device with the possibility to regulate wattage or voltage but without temperature control, or a device with the possibility to regulate wattage or voltage and with temperature control. On certain questions, including this one, participants could also indicate that they did not know the answer. If they used a device with the possibility to regulate wattage or voltage, we asked whether they sometimes increased the wattage during the day and if so, why (e.g., more vapour, warmer vapour), and whether they sometimes reduced the wattage during the day and again if so, why (e.g., less vapour, colder vapour). Current vapers also filled in their favourite flavours, and the brand and type of e-cig and clearomizer they currently used. 

Current non-smoking vapers rated a list of possible complaints they experienced due to vaping (e.g., *throat aches, tendency to cough, bad taste, bad smell*, etc.) on a scale from 1 (*never*), over 3 (*sometimes*), to 5 (*always*). Dependent on participants’ current tobacco status, they indicated to what extent they either were afraid to start smoking tobacco cigarettes, to start smoking tobacco cigarettes again or to completely switch back to tobacco cigarettes again, on a scale from 0 (*not afraid at all*) to 10 (*very afraid*). Current vapers indicated their motivation to stop vaping. 

Subsequently, all participants were asked to indicate what they perceived as most harmful to their health (*tobacco cigarettes, e-cigs, they are both equally harmful to my health, or they are both not harmful to my health*) and to fill in the disadvantages of vaping and smoking (open question). Current vapers answered whether their health had changed since they had started using e-cigs and if so, to fill in what had changed (open question). Then, everyone was asked to rate the harmfulness to their health of tobacco cigarettes, e-cigs, smoking cessation medications, and nicotine replacement therapy (NRT), on a scale from 1 (*not harmful at all*), over 3 (*neutral*), to 5 (*very harmful*). Afterwards, current vapers were asked how much they agreed with several statements, each starting with “Due to using the e-cig...”. The statements reflected potential benefits or improvements (e.g., *quitting smoking, improvements on different health aspects, mood modification*). A scale ranging from 1 (*totally disagree*), over 3 (*neutral*), to 5 (*totally agree*) was used. 

Finally, participants who currently smoked tobacco cigarettes or currently used e-cigs were asked to indicate to what extent they felt dependent on/addicted to nicotine on a scale from 0 (*not dependent/addicted*) to 100 (*very dependent/addicted*). 

### 2.3. Procedure 

Participants were referred to the questionnaire via a weblink. After a short introduction to the study, their consent to participate was asked. When informed consent had been given, the questionnaire was presented. Customers of the online retailer in the Netherlands could win a voucher of 50€ to spend with the online retailer. One voucher was raffled off among them. The study was approved by the Ethics Committee of Thomas More University of Applied Sciences (Ethical code number: TP1718_1220).

### 2.4. Statistical Analyses and Structure of Presentation of Results

Averages (and standard deviations, *SD*s between brackets) and proportions were computed and independent samples t-tests, Chi-square tests, and Wilcoxon rank-sum tests were carried out to explore the differences between the customers of the Dutch online vape shop on the one hand and the Belgian vapers contacted via Facebook on the other hand using SPSS, version 24.0 (IBM, Armonk, NY, USA) [15]. After visual inspection of the data, some outliers were removed, e.g., net income per month of 64,000€, or vaping at 7580 watts.

In what follows, we will describe sociodemographic characteristics from the Dutch Current group, followed by their smoking history and current smoking status, vaping status, reasons for vaping, perceived harmfulness of vaping and smoking, improvements in health and well-being, experienced disadvantages and finally, nicotine dependency. For every theme, we will next make a qualitative comparison of the Dutch Current group and the sample we used in previous research (“Dutch Historical” group) [1] to identify possible evolutions in vaping behaviour over the past two years. Such comparisons were made whenever the Dutch Historical group answered identical or similar questions. Finally, within each theme, the data of the Belgian Current group will be described and quantitatively compared to the Dutch Current group.

## 3. Results

### 3.1. Sociodemographic Characteristics

A small majority of the Dutch Current group were men (around 57%). On average, they were 49 years old and had an average net income of 2338€ per month. The Dutch Historical group was very similar: Most of them were male (57%), on average 46 years old, and they earned on average 2127€ net per month (for more details on sociodemographic characteristics see Table A1 in Appendix A). 

Not surprisingly, with regard to nationality, the Dutch Current group consisted mainly of Dutch participants (96%) and the Belgian Current group consisted mainly of Belgian participants (85%), *χ*² (6) = 326.63, *p* < 0.001. With regard to the other sociodemographic variables, these groups differed in sex, age, and employment status. Both groups contained more men, but the percentage of men was higher in the Belgian Current group, 78% vs. 57%, *χ*² (2) = 22.54, *p* < 0.001. The Belgian Current group was a bit younger, *t*(415) = 7.13, *p* < 0.001 (*M*_Belgian Current_ = 40.9 (10.0), *M*_Dutch Current_ = 48.5 (11.0)). The Belgians were mainly white-collars (40%; 28% blue-collars) whereas the Dutch participants were mainly blue-collars (47%; only 5% white-collars), *χ*² (9) = 71.30, *p* < 0.001. A marginally significant difference was found for income. Net income per month was slightly higher for the Dutch Current group, *t*(95.588) = 1.87, *p* = 0.065 (*M*_Belgian Current_ = 2026 (844), *M*_Dutch Current_ = 2338 (1297)). The groups had a similar educational degree, *χ*² (5) = 7.95, *p* = 0.16.

### 3.2. Smoking History and Smoking Status

All participants in the Dutch Current group had smoked, with all but one, more than 100 tobacco cigarettes. Smokers were on average 16 years (SD = 4.0) old when they had started smoking (for more details on smoking history and smoking status see Table A2 in Appendix A). Only 11% of this sample was currently still smoking, having a rather low cigarette dependency and differing in their motivation to quit. All but three current smokers had made at least one quit attempt. Current smokers reported several experienced disadvantages due to smoking. The most common disadvantages (based on mean rank) were a poor physical condition, worrying about their health, breathing difficulties, and cough tendencies. 

Former smokers had smoked on average 29 years (SD = 12.1) and had needed on average four quit attempts (SD = 1.7) before successfully having quit. More than half of the participants (51%) indicated that they currently used smoking cessation aids. The vast majority of them (97%) specified which cessation aid, which was without any exception the e-cig. The e-cig was judged to be the most effective smoking cessation aid. All participants who had quit smoking or who indicated that they had made at least one quit attempt answered how long they had quit with their most successful attempt. This was on average 35.9 months (SD = 38.1, range = 0.04–240).

Again, many similarities can be noted with the Dutch Historical group: 99% of them had smoked in the past, on average had started smoking when they were 15, had tried to quit five times, and had made use of different cessation methods while reporting that the e-cigarette was the most effective. In this Dutch Historical group, 17% was still smoking at the time of the study (a slightly higher percentage than in our Dutch Current group), and those current smokers reported mainly experiencing the same complaints due to smoking.

With regard to smoking tobacco cigarettes, only a few differences between the Dutch and Belgian Current group were found. They differed in the use of smoking cessation aids, in that 61% of the Belgian Current group reported to currently using cessations aids vs. 51% of the Dutch Current group, *χ*² (1) = 4.14, *p* < 0.05. The Belgian participants who had quit smoking or had made at least one quit attempt in the past had used more smoking cessation aids, *t*(371.771) = 2.42, *p* < 0.01 (*M*_Belgian Current_ = 2.6 (1.7), *M*_Dutch Current_ = 2.2 (1.3)), although there was no difference between the groups with regard to the number of *effective* smoking cessation aids. Former smokers of the Dutch Current group had smoked longer than former smokers of the Belgian Current group, *t*(264.415) = 5.01, *p* < 0.001 (*M*_Belgian Current_ = 22.7 (10.2), *M*_Dutch Current_ = 28.6 (12.1)). Current smokers of the Belgian Current group were more motivated to quit than current smokers of the Dutch Current group, *t*(42) = 2.94, *p* < 0.01 (*M*_Belgian Current_ = 4.6 (1.7), *M*_Dutch Current_ = 3.1 (1.6)).

### 3.3. Vaping Status and Technical Aspects

Almost all (97%) participants of the Dutch Current group had already tried an e-cig, and 99% of those vaped every day, see Table 1 for more details on vaping history and current vaping status. More than half of them did not want to quit vaping. Current vapers had been using the e-cig for more than three years. These figures again resemble those of the Dutch Historical group very closely: 96% had already tried an e-cigarette, 97% of those were regular (daily/weekly) vapers, having used the e-cigarette for more than two years.

The participants of the Belgian Current group were even less motivated to quit vaping than those of the Dutch Current group, *t*(408) = 2.87, *p* < 0.01, but had been using the e-cig for a shorter length of time than the participants of the Dutch Current group, *t*(407) = 0.18, *p* < 0.01, see Table 1. 

Two-thirds of the Dutch Current vapers used mouth-to-lung inhalation vs. only 17% who used direct lung inhalation and 16% who used both inhalation techniques. The majority seldom or never experienced a dry hit, see Table 1. Answers to the open question as to what they did to prevent dry hits varied but most answers referred to ‘refill the tank in due time’ to ensure a constant e-liquid supply to the coil. Remarkably, the majority of Belgians vapers usually inhaled directly into the lungs, whereas most Dutch vapers usually inhaled mouth-to-lung, *χ*² (2) = 111.93, *p* < 0.001, see Table 1. Participants of the Dutch Current group were less familiar with the concept of a dry hit, *χ*² (3) = 57.90, *p* < 0.001, see Table 1.

On average, in the Dutch Current group, regular vapers used 21.8 mL e-liquid per week (SD = 15.2), with an average nicotine concentration of 8.9 mg/mL (SD = 5.3), see Table 2 for more details on vaping parameters. The Dutch Historical group consumed a comparable amount of e-liquid (19.5 mL/week) with almost the same nicotine concentration (9.7 mg/mL). The participants of the Belgian Current group used e-liquid with a nicotine concentration that was almost three-times lower, *t*(226.972) = 11.13, *p* < 0.001 (*M* = 3.3 mg/mL (3.9)); however, they also used almost triple the amount of e-liquid per week than the Dutch Current group, *t*(338.661) = 12.10, *p* < 0.001 (*M* = 62.9 (51.0)). 

The majority of the Dutch Current group (59%, vs. 32% of the Belgian Current group) used the same amount of e-liquid as when they had started to vape. Only 17% of the participants of the Dutch Current group, vs. 49% of the Belgian Current group, used more e-liquid now, *χ*² (2) = 44.87, *p* < 0.001. In both groups, the majority of the participants indicated that they had decreased the nicotine concentration since they had started to vape; however, this percentage was higher in the Belgian Current group (91% vs. 70%), *χ*² (2) = 29.30, *p* < 0.001. Only a few persons in both groups indicated that they had increased the nicotine level since they had started to vape (2% in the Belgian Current group and 3% in the Dutch Current group). 

The groups differed with regard to the PG/VG ratio, with the Dutch Current group being associated with a higher propylene glycol amount (*M*_PG/VG_ = 63.5/36.5), and the Belgian Current group using more vegetable glycerine (*M*_PG/VG_ = 36.8/63.2), *t*(356.015) = 16.18, *p* < 0.001.

Around three-quarters of the Dutch Current vapers had one favourite flavour (in the Dutch Historical group this was 94%), whereas only 26% of the Belgian Current group had one favourite flavour, *χ*² (1) = 93.65, *p* < 0.001. The most frequently mentioned flavours in the current samples were tobacco (31%), fruit flavours (18%), varied flavours such as Red Astaire or Heisenberg (17%), and mint/menthol (13%). Tobacco (55%), fruit (14%), and mint (11%) were the most-liked flavours in the Dutch Historical group. 

The mean coil resistance used by the Dutch Current group was 1.3 ohm (SD = 0.7), which was significantly higher than in the Belgian Current group, *t*(149.647) = 13.54, *p* < 0.001 (*M* = 0.4 ohm (0.4)). Around 10% of the participants of both groups indicated that they did not know which resistance they usually used.

The mean power used by the Dutch Current group was 26.6 W (SD = 36.9) while the Belgian Current group usually vaped at more than double this wattage, *t*(333) = 6.84, *p* < 0.001 (*M* = 61.0 (40.4)). Around 20% of the participants of both groups indicated that they did not know at what wattage they usually vaped. Voltage was then assessed in order to be able to calculate wattage; however, 73% of the participants who could not fill in the wattage did not know the voltage they used either. 

The majority of the Dutch Current group had not changed the wattage since they had started to vape (81%, vs. 23% of the Belgian Current group), *χ*² (2) = 132.19, *p* < 0.001. The percentage of the Belgian Current group that had increased the wattage since they had started to vape was much higher than the percentage of the Dutch Current group who had done so (66% vs. 12%). The most common reasons to increase the wattage were to obtain more vapour (46%) or ‘because I currently use an e-liquid with less nicotine’ (38%). The most common reason to decrease the wattage was to obtain a colder vapour (43%).

Because of the generality and/or uncertainty of the answers, it was not possible to get a detailed picture of how many times per day vapers used their e-cig, of how many puffs they took per vaping bout, of how many puffs they took per day, and of the exact brand and type of e-cig and clearomizer they used. Regarding the type of device, there were significant differences between groups, *χ*² (3) = 83.78, *p* < 0.001: Of the Dutch Current vapers, around 17% had a device with the possibility to regulate wattage or voltage with temperature control (vs. 61% of the Belgian Current group), 41% had a device with the possibility to regulate wattage or voltage but no temperature control (vs. 23% of the Belgian Current group), 35% had no possibility to regulate wattage/voltage/temperature (vs. 16% of the Belgian Current group), and 6% of the Dutch Current participants did not know. Vapers of the Belgian Current group, with the possibility to regulate wattage or voltage, did so more often than the Dutch Current vapers did; 30% of the Belgians Current vapers indicated that they sometimes increased the wattage during the day (vs. 13% of the Dutch Current group, usually to obtain more vapour, a warmer vapour or a stronger nicotine hit), *χ*² (1) = 8.26, *p* < 0.01. One-third (33%) of the Belgian Current group indicated that they sometimes decreased the wattage during the day (vs. 15% of the Dutch Current group, usually to obtain less vapour, a colder vapour or a weaker nicotine hit), *χ*² (1) = 9.68, *p* < 0.01. 

### 3.4. Reasons for Vaping

The most common reasons to start vaping given by both current groups were to quit smoking and because it is healthier than tobacco cigarettes. These were also the most common reasons to continue vaping. An important additional reason to continue vaping was ‘because it is tasty and I enjoy it’. For other reasons to start and continue vaping, we refer to Table 3. 

There were no statistically reliable differences between the Dutch Current and the Dutch Historical group. There were some differences between the Dutch Current group and the Belgian Current group with regard to their reasons to start vaping. Financial reasons, *χ*² (1) = 13.74, *p* < 0.001, and ‘because smoking is prohibited in certain places’, *χ*² (1) = 12.35, *p* < 0.001, were indicated more frequently as reasons to start vaping by the Dutch Current group than by the Belgian Current group. To smoke less, *χ*² (1) = 5.83, *p* < 0.05, and different flavours, *χ*² (1) = 4.84, *p* < 0.05, were indicated more often by the Belgian Current group than by the Dutch Current group. There were also some differences between the Dutch Current group and the Belgian Current group with regard to their reasons to continue vaping. Financial reasons and ‘because smoking is prohibited in certain places’ were again indicated more frequently by the Dutch Current group than by the Belgian Current group, *χ*² (1) = 13.94, *p* < 0.001 and *χ*² (1) = 11.16, *p* < 0.001, respectively. Even though it was very low in absolute numbers, the Dutch Current group also indicated more often that they continue vaping out of curiosity, *χ*² (1) = 4.407, *p* < 0.05. The Belgian group indicated more often that they continue vaping because of the different flavours, *χ*² (1) = 27.78, *p* < 0.001, to pass time, *χ*² (1) = 11.44, *p* < 0.01, and to quit smoking, *χ*² (1) = 4.14, *p* < 0.05. 

### 3.5. Perceived Harmfulness

When Dutch Current participants were asked to indicate what they perceived as most harmful to their health (tobacco cigarettes, e-cigs, they are both equally harmful to my health, or they are both not harmful to my health), the large majority of the participants (94%) chose tobacco cigarettes. Only one participant indicated that e-cigs are more harmful to health. Eight participants (5%) believed that they are both equally harmful, and nobody indicated that they are both not harmful. In the Dutch Historical group, the same amount (94%) perceived tobacco cigarettes as the most harmful and a few (6%) believed that both were equally harmful. The Belgian Current group answered very similarly, only proportionally less participants of the Belgian Current group perceived both as equally harmful (2% vs. 5%), *χ*² (2) = 7.17, *p* < 0.05.

We also asked about the perceived harmfulness of smoking, vaping, smoking cessation medications, and nicotine replacement therapy, see Table 4. Smoking was considered harmful to very harmful by 99% of the Dutch Current group, whereas almost 25% considered vaping to be harmful to very harmful. Vaping was considered equally harmful as other cessation methods and nicotine replacement therapy. This pattern of answers was almost identical to that of the Dutch Historical group. A few differences between both current groups were found: The Belgian Current group considered vaping less harmful than the Dutch Current group, Ws = 48677.0, *z* = −8.48, *p* < 0.001 (*Mdn*_Belgian Current_ = 2, *Mdn*_Dutch Current_ = 3), but perceived other cessation methods as more harmful than the Dutch Current group, Ws = 27490.5, *z* = −3.88, *p* < 0.001 (*Mdn*_Belgian Current_ = 3, *Mdn*_Dutch Current_ = 3). The groups did not differ in how harmful they considered smoking, Ws = 31269.0, *z* = −0.72, *p* = 0.470 and nicotine replacement therapy, Ws = 30864.0, *z* = −0.89, *p* = 0.371.

### 3.6. Improvements in Health and Well-Being and Experienced Disadvantages

Around 80% of the Dutch Current vapers indicated that their health had changed since they used the e-cig. The most frequently reported answers to the open question as to what had changed were: A better condition (one participant answered, “I think my condition has deteriorated”, though), better breathing, less coughing, improved smell and taste, and less illness. Other answers were, amongst others: Better sleep, healthier gums, teeth and hair, and no more stinking. A few participants reported a small weight gain, but also a loss of weight was mentioned. Both the percentage (84% in the Dutch Historical group) and the reported changes in health were in line with those given by the Dutch Historical group.

As can be seen in Table 5, a great majority of the vapers agreed to totally agreed with several possible improvements due to using e-cigs; again, a very similar picture was seen in the Dutch Historical group. Current Dutch vapers largely agreed, for example, that their craving for a tobacco cigarette had decreased, that they could decrease their smoking consumption, that they could quit smoking, that they had fresher breath, and that their physical condition and health had improved. ‘My quality of sleep has improved’, ‘I am in a better mood’, and ‘I can vape in more contexts’ were the only three possible improvements with which less than half of the participants agreed, but even then only a minority disagreed. With regard to having more technical issues, the opinions were diverse.

More participants of the Belgian Current group answered affirmatively to the question of whether their health had changed since they started to vape (94% vs. 80%), *χ*² (1) = 20.12, *p* < 0.001. Many differences between both Current groups were found with regard to improvements in health, see Table 5. The Dutch Current group scored higher than the Belgian Current group on most improvements. However, the Belgian Current group scored higher on ‘I bother bystanders less’ and ‘I can vape in more contexts’. 

Participants seemed to experience a few disadvantages due to vaping. The most common disadvantages (based on mean rank) reported by current non-smoking vapers were a dry mouth, dry throat, and increased weight. An overview of all disadvantages with the distributions of the responses can be found in Table 6. Although both Current groups indicated that they experienced few disadvantages due to vaping, many differences between groups were found, see Table 6. The Belgian Current group suffered more from a dry mouth and a bad taste than the Dutch Current group. The Dutch Current group suffered more from a bad taste upon inhaling, an unpleasant sensation in the throat when inhaling, worrying about their health, sleeping difficulties, bad physical condition, increased heart rate or palpitations, throat ache, breathing difficulties, coughing up mucous, and a bad sense of smell than the Belgian Current group.

### 3.7. Nicotine Dependency

Participants who currently smoked tobacco cigarettes or currently used e-cigs were asked to indicate to which extent they felt dependent on /addicted to nicotine. The Dutch Current group felt more dependent on nicotine than the Belgian Current group, *t*(407) = 7.68, *p* < 0.001 (*M*_Dutch Current_ = 64.8 (27.9); *M*_Belgian Current_ = 41.9 (29.4)), see Table 7.

Finally, all participants indicated to what extent they were afraid to start smoking tobacco cigarettes (again) or to switch completely back to tobacco cigarettes again, by moving a slider on a scale from 0 (*not afraid at all*) to 10 (*very afraid*). Current vapers with a smoking history were only a little bit afraid to start smoking tobacco cigarettes again (*M*_Dutch Current_ = 1.6 (2.5); *M*_Belgian Current_ = 1.7 (2.6)), with no differences between groups, *t*(309) = −0.57, *p* = 0.74. Dutch Current dual users were less afraid to switch completely back to tobacco cigarettes again (*M* = 3 (2.2)) than Belgian Current dual users (*M* = 4.2 (3.4)), *t*(47) = −1.34, *p* < 0.05.

## 4. Discussion

In this study, we found that the background smoking profile of all current participants, as well as their reasons for starting vaping, the many experienced improvements in health and well-being and the few disadvantages listed, were very similar to the results we observed in our previous study [1]. Results were also in line with data gathered through surveys of brick-and-mortar vape shop customers [2,3] and data from convenience sampling of vapers who visited, for example, e-cigarette discussion fora or smoking cessation websites [4,5,6,7]. 

For the Current Dutch and Current Belgian samples combined, all (but one) participants had smoked before, and almost all had smoked more than 100 tobacco cigarettes. Only 10% of the total sample was currently still smoking. Almost all (99%) participants had already tried an e-cig, and 96% of those vaped every day. A great majority of the vapers (89%) indicated that their health had changed since they used the e-cig and agreed to totally agreed with several possible improvements due to using e-cigs. Smoking was considered harmful to very harmful by almost everyone, whereas the majority considered vaping not harmful at all to not harmful. 

Results showed that participants of the Dutch Current group resembled those of the Dutch Historical group very well. Contrastingly, many differences between the participants of the Dutch Current group and Belgian Current group were found, mostly related to the more technical aspects of vaping; we will now take a closer look at the most striking dissimilarities.

The Belgian Current group used a lower coil resistance (0.4 vs. 1.3 Ohm) and a higher wattage (61 vs. 26.6 W) compared to the Dutch Current group. Almost two out of three Belgian Current vapers had increased the wattage since they had started to vape (vs. only 12% in the Dutch Current group), mainly to obtain more vapour or a stronger nicotine hit. In line with vaping at a higher wattage, the Belgian Current group used almost three times the amount of e-liquid (62.9 mL) per week as the Dutch Current group (21.8 mL). However, the Belgian Current group used e-liquid with a much lower nicotine concentration (3.3 mg/mL) than the Dutch Current group (8.9 mg/mL). These data are in line with those of the previously mentioned pilot study [8] in which 74% of questioned vapers used e-liquid with a nicotine concentration lower than 4 mg/mL and 40% used more than 70 mL per week.

When we multiply the volume of e-liquid with the nicotine concentration (mL liquid * mg/mL), we get comparable results for the total weekly nicotine consumption: 208 mg for the Belgian Current group, which is a bit higher than the 194 mg for the Dutch Current group (vs. 189 mg in the Dutch Historical group). Interestingly, this is comparable with the nicotine intake measured in cigarette smokers; for example, in a study by Benowitz and Jacob, the average weekly nicotine intake was 263.2 mg [16]. Research showed that both in vapers [17] and in tobacco smokers [18], the retention rate of nicotine (% of the consumed dose) is around 99%, at least when the e-cigarette aerosol or the tobacco smoke is inhaled. These observations again confirm the validity of the notion that nicotine users tend to self-titrate nicotine up to the serum blood levels they are used to, a phenomenon which has also been described in the context of cigarette smokers switching to cigarettes with a lower than usual nicotine content (“light cigarettes”) and smoking their cigarettes more intensively to compensate for this lower nicotine content [19]. 

Half of the Belgian Current group (49%) indicated that they used more e-liquid now (vs. only 17% in the Dutch Current group) and around 9 out 10 (vs. 7 out of 10 in the Dutch Current group) had decreased their nicotine concentration since they had started to vape. Similar patterns were also found in a longitudinal study on the cotinine level (a metabolite and biomarker of nicotine) in long-term daily e-cig users by Etter [20], and more recently, in a study by Soar and colleagues [21]. Etter [20] asked 98 exclusive vapers to send in saliva vials at the beginning of the study, as well as eight months later. The cotinine level in this saliva was analyzed, and data showed no change over time. Vapers also recorded the average amount of e-liquid and nicotine concentration consumed. The median nicotine concentration decreased significantly from 11 mg/mL at baseline to 6 mg/mL at follow-up, while median liquid volume increased from 80 mL per month at baseline to 100 mL at follow-up. In a 12-month prospective study, Soar and colleagues followed 27 vapers who had been vaping for on average three years [21]. Results again showed no change over time in salivary cotinine levels, indicating invariant nicotine consumption, while a significant reduction in the nicotine concentration was reported, together with a significantly higher volume of e-liquid consumed. 

The latter finding suggests similar systemic absorption of nicotine from bigger volumes of low-nicotine aerosol than from smaller volumes of high-nicotine aerosol. To the extent that the (other) HPHCs of the aerosol would show retention rates similar to that of nicotine, consuming bigger volumes of low-nicotine liquid/vapour would also result in systemic absorption of larger quantities of HPHCs. Even though studies on the pharmacokinetics and retention rates of HPHCs in e-cig aerosols resulting from high-wattage-large-volume vs. low-wattage-small-volume vaping are currently largely lacking, some recent data do suggest that this may indeed be the case. For example, Dawkins and colleagues [12] conducted a 4-week “real-world” counterbalanced cross-over study manipulating nicotine concentration (low, 6 mg/mL vs. high, 18 mg/mL) and power settings (fixed vs. adjustable wattage). Results showed that providing participants with low-nicotine e-liquid resulted not only in reliably more liquid consumption compared with the high-nicotine conditions, but, as measured by the amount of urinary formate, in more absorption of the carcinogen formaldehyde (particularly in the adjustable wattage conditions). Interestingly, in our study, the Belgian Current group scored significantly lower than the Dutch Current group regarding the most reported improvements in health, which might be related to their higher liquid consumption and potentially higher HPHCs absorption. However, until future research indicates a clear link, this hypothesis remains largely speculative. For example, the Dutch Current vapers were also older and, therefore, probably had started to vape while already having more smoking-related co-morbidities (e.g., bronchitis, breathlessness). This, rather than the hypothesized lower exposure to HPHCs, could also explain why they reported to have improved more health wise. 

Some possible explanations for the observation that some people deliberately choose an e-liquid with a low nicotine concentration or try to further reduce this concentration can be offered. Epidemiological data on the health effects of long-term use of a related smokeless low-risk product delivering high levels of nicotine, Swedish snus, as well as on the health effects of classical nicotine replacement therapies, show that long-term use of nicotine per se does not result in any measurable increase in the risk of serious cardiovascular or respiratory diseases, nor in a higher cancer risk [22,23,24,25]. In sum, speaking strictly from a health perspective, there is little that would justify a choice for low-nicotine e-liquid or a desire to further lower nicotine concentrations, at least not at the doses typically consumed by vapers [26,27]. However, the desire to minimize nicotine intake may be instigated by other causes that are related to a fear of nicotine that is nourished by several societal influences. For example, as suggested earlier [21], an upper limit of 20 mg/mL nicotine has been imposed on the sale of e-liquids by the EU Tobacco Products Directive (TPD) in May 2016. Such legislation adds to the perception that nicotine in itself might be a (or even *the*) particularly dangerous and unhealthy (and thus to be avoided) component of vaping. Also, in some health professionals, nicotine is perceived as “the bad guy” in smoking and vaping alike [28], and communication of this belief to smokers may further enhance the fear of nicotine. As we argued earlier [28], this misconception can pose a serious threat to the success of Tobacco Harm Reduction (THR) [29] strategies and it could also be the reason why e-liquids with a low nicotine concentration are preferred by many in the current sample. Apart from being motivated by a fear of nicotine, the choice for low(er) nicotine concentrations may also, however, be caused by more mundane factors. In recent years, high-wattage devices have become widely available, and correspondingly, e-liquids with very low nicotine concentrations have become the rule rather than the exception in the standard market supply. Remarkably, and largely in parallel with the “low nicotine high wattage vaping” trend, a diametrically opposite development is also currently taking place in countries outside the TPD—mainly so in the USA. Since the introduction of the JUUL "pod"-based e-cigarette [30], low-wattage vaping (lower than 10 watts) of e-liquids with very high concentrations (50 mg/mL or more) of protonated nicotine (containing “nicotine salts”) has, starting in 2016, taken a large share of the closed-system e-cigarette market in the USA. TPD forbids the marketing of these devices (and/or high-nicotine e-liquids) in the European market, even though “down-tuned” pod-based systems and e-liquids with nicotine salts at TPD-compliant levels of 20 mg/mL have recently been introduced to the Belgian and Dutch markets.

Finally, some testable hypotheses may be advanced with respect to the causes of the difference in the technical aspects of the vaping behaviour between the Dutch Current and Belgian Current samples. The explanatory candidates that may be the focus of future research include (1) differential product availability (e.g., low/high nicotine concentration e-liquids, low/high wattage devices, mouth-to-lung vs. direct-lung devices); (2) differential product recommendations to (new) customers by Belgian vape-shop owners vs. Dutch (online) vape shops; (3) sociodemographic differences between populations of vapers (e.g., age, employment, era when started to vape); (4) differential personal beliefs about and attitudes towards vaping and nicotine use; and (5) differential vaping subcultures and vaper identities (e.g., “cloud chasers”/”substitute(rs)” [31].

This study has some limitations. First, we have included some new questions that have not been asked in the Dutch Historical Group, implying that we cannot make a qualitative comparison for each question. At the time of our previous research on the typical profile of customers of online vape stores, devices were not yet so advanced and certain vaping trends had not yet been observed. Secondly, our data rely on self-reporting, but that was the case in all groups that we compared, so if errors did occur, they should occur in all groups. Finally, it goes without saying that the results obtained here should not be generalized to the total population of Dutch/Belgian vapers because of a selection bias for having positive experiences with and being enthusiastic about vaping is highly likely in the samples described here.

## 5. Conclusions

In this study, we found that certain evolutions can be discerned with regards to the more technical aspects of vaping, such as vaping at high wattage, using e-liquids with low nicotine concentrations, but consuming big volumes of such e-liquids. However, these trends are clearly not observed among all vapers, but rather, appear to reflect the existence of certain subcultures or regional differences. The potential health effects of this trend remain largely unexplored but do merit closer attention and future investigation.

## Figures and Tables

**Figure 1 ijerph-16-00723-f001:**
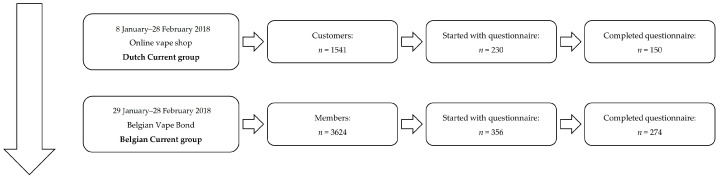
Flowchart of recruitment of participants.

**Table 1 ijerph-16-00723-t001:** Vaping history and current vaping status for the Dutch Current/Belgian Current group.

Variable	Dutch Current Group		Belgian Current Group	
	*n*	*M (SD)* or %	*n*	*M (SD)* or %
**Ever Vaped**	150		274	
Yes		96.7		99.6
**Vaping frequency (in ever vapers)**	145		273	
Every day		98.6		94.5
Some days a week		1.3		2.6
Former vapers		0.0		2.9
Months vaping (in ever vapers)		46.8 (21.8)		37.1 (35.6)
**Motivation to quit vaping**	145		265	
I don’t want to quit		33.8		55.8
I think it is better to quit, but I don’t want to		28.3		14.0
I want to quit, but I don’t know when		26.9		19.2
I really want to quit, but I don’t know when		7.6		9.8
I want to quit within the next 3 months		2.8		1.1
I want to quit within the next month		0.7		0.0
I want to quit now		0.0		0.0
**Inhalation**	145		265	
Usually mouth-to-lung inhalation		66.9		16.2
Usually direct lung inhalation		17.2		60.8
Both		15.9		23.0
**Dry hit**	145		265	
Never		8.3		6.8
Seldom		54.5		83.4
Regularly		5.5		4.9
I don’t know what a dry hit is		31.7		4.9

**Table 2 ijerph-16-00723-t002:** Vaping parameters of the Dutch Current/Belgian Current group.

Variable	Dutch Current Group		Belgian Current Group	
	*n*	*M* (*SD*) or %	*n*	*M* (*SD*) or %
Volume of e-liquid (mL/week)	141	21.8 (15.2)	263	62.9 (51.0)
Nicotine concentration (mg/mL)	143	8.9 (5.3)	263	3.3 (3.9)
PG/VG ratio	145	63.5/36.5	265	36.8/63.2
Coil resistance (ohm)	116	1.3 (0.7)	249	0.4 (0.4)
Power (W)	82	26.6 (36.9)	253	61.0 (40.4)
Type of device	145		264	
with possibility to regulate wattage or voltage with temperature control		16.6		61.0
with possibility to regulate wattage or voltage but no temperature control		41.4		22.7
no possibility to regulate wattage/voltage/ temperature		35.9		16.3
I don’t know		6.2		0.0

Note. PG = propylene glycol, VG = vegetable glycol.

**Table 3 ijerph-16-00723-t003:** Distribution of responses (in %) regarding the reasons for starting to use e-cigs (all participants who had ever used an e-cig) and the reasons for continuing to use e-cigs (all current vapers) for the Dutch Current/Belgian Current group.

Reasons for Vaping	Dutch Current Group		Belgian Current Group	
	Start (*n* = 145)	Continue (*n* = 145)	Start (*n* = 273)	Continue (*n* = 265)
To quit smoking	74.7	39.3	81.8	49.6
Because it is healthier than tobacco cigarettes	54.0	44.0	48.2	44.9
Financial reasons	32.0	20.0	16.4	7.7
To smoke less	12.7	8.7	22.3	5.8
Out of curiosity	15.3	2.7	16.8	0.4
Different flavours	7.3	5.3	14.6	26.3
Other (e.g., reduce nicotine intake, pregnancy)	6.7	15.3	6.9	10.9
Because smoking is prohibited in certain places	12.0	8.7	3.3	1.8
Others offered it	6.0	0.0	6.2	0.4
To pass time	4.0	7.3	2.2	19.7
Dual use	2.0	2.0	2.9	2.2
Others do it	0.7	0.0	0.4	0.0
Because it is tasty and I enjoy it		44.7		42.0

**Table 4 ijerph-16-00723-t004:** Distributions of responses (in %) of perceived harmfulness for smoking, vaping, smoking cessation medications, and nicotine replacement therapy (NRT) for the Dutch Current/Belgian Current group.

Variable	Not Harmful at All	Not That Harmful	Neutral	Harmful	Very Harmful
Smoking	0.0/0.04	0.0/0.0	1.3/1.1	20.7/17.5	78.0/81.0
Vaping	7.3/31.8	40.0/52.6	28.0/9.9	22.7/4.7	2.0/1.1
Smoking cessation medications	7.3/2.6	14.0/11.7	50.0/39.1	27.3/39. 1	1.3/7.7
NRT	7.3/5.8	24.0/21.9	45.3/47.1	21.3/20.1	2.0/5.1

**Table 5 ijerph-16-00723-t005:** Distributions of responses (in %) by current vapers (*n* = 418) on a list of several possible improvements due to using e-cigs and differences between customers of the Dutch online vape shop and Belgian vapers contacted via Facebook with regard to these possible improvements.

Variable	Totally Disagree	Disagree	Neutral	Agree	Totally Agree	Ws	*z*	*Mdn Belgian*	*MdnDutch*
My craving for a tobacco cigarette has decreased	0.5	1.4	3.8	11.2	83.0	54,861.0 **	−3.04	1	1
I could decrease my smoking consumption	2.2	1.9	5.7	12.2	78.0	53,818.0 ***	−3.97	1	1
I could quit smoking	1.9	4.5	5.5	6.7	81.3	53,887.5 ***	−4.14	1	1
I have fresher breath	0.2	0.7	8.1	19.1	71.8	53,664.0 ***	−3.80	1	1
My physical condition and health has improved	1.0	1.2	8.9	16.7	72.2	54,289.5 **	−3.14	1	1
I can breathe better	0.5	2.2	12.2	21.8	63.4	51,672.5 ***	−5.49	1	2
I gain more pleasure from vaping than from smoking	1.4	3.8	14.8	14.6	65.3	49,844.0 ***	−7.39	1	2
I have improved smell	0.2	2.2	17.9	20.1	59.6	52,220.0 ***	−4.81	1	2
I have improved taste	0.5	2.2	17.2	22.5	57.7	51,649.5 ***	−5.30	1	2
I bother bystanders less	2.9	7.9	20.8	23.0	45.5	24,348.0 ***	−5.45	2	1
My appetite has improved	0.2	4.8	42.6	19.9	32.5	53,479.5 **	−3.37	2	3
My quality of sleep has improved	1.4	5.7	46.2	19.4	27.3	50,710.0 ***	−5.90	2	3
I am in a better mood	1.2	4.8	50.5	20.3	23.2	53,882.0 **	−3.06	3	3
I can vape in more contexts	12.2	16.0	23.7	16.7	31.3	21,742.0 ***	−7.56	3	1
I have more technical issues	13.2	27.0	31.8	18.7	9.3	28,920.0	−1.28	3	3

Note. ** *p* < 0.01, *** *p* < 0.001.

**Table 6 ijerph-16-00723-t006:** Distributions of responses (in %) of current non-smoking vapers (*n* = 369) for several experienced disadvantages due to vaping and differences between vapers of the Dutch Current vs. Belgian Current group with regard to these disadvantages.

Variable	Never	Seldom	Sometimes	Often	Always	Ws	*z*	*MdnBelgian*	*MdnDutch*
Dry mouth	36.9	28.2	27.6	6.5	0.8	25,919.5 **	−3.31	2	1.5
Dry throat	46.6	30.1	20.6	2.4	0.3	28,059.0	−1.37	2	1
Increased weight	64.2	14.4	11.7	8.1	1.6	53,675.0	−0.66	1	1
Cough tendencies	54.2	36.0	8.9	0.8	0.0	53,839.5	−0.47	1	1
Bad taste upon inhaling	50.7	41.5	7.9	0.0	0.0	52,087.5 *	−2.20	1	2
Unpleasant sensation in throat when inhaling	58.3	32.8	8.1	0.3	0.5	51,247.0 **	−3.07	1	2
Bad taste (“vaper’s tongue”)	64.0	25.5	9.8	0.8	0.0	25,499.5 ***	−4.14	1	1
Worrying about my health	74.0	14.6	8.9	2.4	0.0	51,612.0 **	−2.98	1	1
Sleeping difficulties	78.9	12.5	4.9	3.3	0.5	51,867.0 **	−2.93	1	1
Headache	74.3	17.3	7.3	0.8	0.3	53,726.0	−0.68	1	1
Bad physical condition	80.5	12.7	3.8	2.2	0.8	51,546.5 **	−3.29	1	1
Increased heart rate or palpitations	79.7	14.1	6.0	0.3	0.0	52,056.0 **	−2.72	1	1
Throat ache	75.6	20.6	3.5	0.3	0.0	52,335.5 *	−2.31	1	1
Breathing difficulties (shortness of breath, breathlessness)	79.4	14.4	5.1	1.1	0.0	51,141.0 ***	−3.88	1	1
Coughing up slimes	79.4	14.4	5.7	0.5	0.0	52,316.0 *	−2.45	1	1
Unpleasant odors	87.0	10.6	2.2	0.3	0.0	53,084.5	−1.81	1	1
Bad sense of smell	89.4	8.4	1.1	1.1	0.0	52,493.0 **	−2.850	1	1
Bronchitis	90.5	8.4	0.5	0.3	0.3	28,512.0	−1.72	1	1

Note. * *p* < 0.05, ** *p* < 0.01, *** *p* < 0.001.

**Table 7 ijerph-16-00723-t007:** Nicotine dependency and fear to start smoking again for the Dutch Current/Belgian Current group.

Variable	Dutch Current Group		Belgian Current Group	
	*n*	*M (SD)* or %	*n*	*M (SD)* or %
Nicotine dependency	145	64.8 (27.9)	264	41.9 (29.4)
Afraid to start smoking again or to completely switch back to tobacco cigarettes				
Current vapers with a smoking history	96	1.6 (2.5)	215	1.7 (2.6)
Dual users	18	3.0 (2.2)	31	4.2 (3.4)

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
