# Peer review of "When Less is More: Vaping Low-Nicotine vs. High-Nicotine E-Liquid is Compensated by Increased Wattage and Higher Liquid Consumption"

_ijerph, 2019, doi:10.3390/ijerph16050723_

Round 1

Reviewer 1 Report

The present manuscript is methodologically correct, it's understandable and raises a topic of interest at the level of public health.

A comment in relation to the hypothesis about the differences between the two groups studied. The authors raise differences that could be due especially to personal factors, we do not know if at a health level the campaigns are similar or the environment recommendations have been similar.

It is known but still surprising the great difficulty that smokers have to completely quit smoking, and they smoke new products, e-cigarette, with the belief that this will help them to quit smoking, although they are  mostly precontemplators or contemplators who do not raise quit smoking (motivation to quit vaping). 

In the discussion it would have been interesting to focus this problem more clearly and to propose future lines of prevention and health promotion with these results.

Author Response

The present manuscript is methodologically correct, it's understandable and raises a topic of interest at the level of public health.

è We would like to thank the Reviewer for the careful reading of our manuscript and the positive evaluation.

A comment in relation to the hypothesis about the differences between the two groups studied. The authors raise differences that could be due especially to personal factors, we do not know if at a health level, the campaigns are similar or the environment recommendations have been similar.

è We thank the reviewer for his/her suggestion. However, we do not think it is the case that differences between the Dutch and Belgian samples can be caused by differences in "regulatory context" and/or recommendations/positions of health institutions. First, the Tobacco Products Directive (TPD) was implemented almost at the same time in The Netherlands as in Belgium. Second, in both countries a "cautiously positive" approach is implemented: e.g., only for smokers who want to quit smoking, vaping is promoted because it is less harmful; full switch is needed; it is best to stop vaping afterwards, because long-term effects unprecedented; never ok for non-smokers/young people; fear of gateway & renormalization.

It is known but still surprising the great difficulty that smokers have to completely quit smoking, and they smoke new products, e-cigarette, with the belief that this will help them to quit smoking, although they are  mostly precontemplators or contemplators who do not raise quit smoking (motivation to quit vaping). In the discussion it would have been interesting to focus this problem more clearly and to propose future lines of prevention and health promotion with these results.

è We agree with the Reviewer that working on motivation is a valuable strategy, however, the aim of this paper was to compare vaping behavior between two groups/timeframes and therefore this issue lies beyond the scope of this manuscript.

Reviewer 2 Report

General:

Overall, the manuscript is well written and provides much useful information regarding the evolution of ECIG usage and subcultural/geographical differences. Because this investigation presents so much data, the manuscript would greatly benefit from the addition of four more tables (indicated below) comparing groups of participants (i.e. Dutch historical, Dutch current and Belgian current). This would allow the reader to see the information and make comparisons quickly.

Abstract:

Line 34 states that “the Belgian vapers probably expose themselves to larger quantities of harmful or potentially harmful constituents (HPHCs)”. The word probably is too strong a word and should be replaced with “may have a greater potential to” (or something similar). This would be more in line with what the author’s state later in the manuscript, “until future research indicates a clear link, this hypothesis remains largely speculative” (Line 500).

Methods:

A flowchart showing timelines and procedures used to recruit participants would be helpful.

Results:

Statistical Analyses and Structure of Presentation of Results (Lines 173 to 188) needs to be included as part of the methods, not the results.

A table showing sociodemographics for Dutch historical (when data is available), Dutch current and Belgian current should be provided for ease of comparison.

A table showing smoking history and status for Dutch historical (when data is available), Dutch current and Belgian current should be provided for ease of comparison.

A table showing vaping parameters (i.e. volume of E-liquid consumed, nicotine used PG/VG ratio used, types of devices used (to include wattage, voltage, temperature, resistance of coils and perhaps a list of the most common flavors preferred)  for Dutch historical (when data is available), Dutch current and Belgian current should be provided for ease of comparison. If available, it would be nice to know the type of coils preferred (i.e. kanthal, nichrome, stainless steel, etc).

A table showing nicotine dependency for Dutch historical (when data is available), Dutch current and Belgian current should be provided for ease of comparison.

The already existing tables should also include data from the Dutch historical group (when data is available)

Discussion:

The discussion is long and should be shortened. There is no need to summarize all the results of the present study. Only mention the results required to argue a point (either for or against) with what is already available in the primary literature or to propose a new idea or concept based on the findings of the study.

Author Response

General:

Overall, the manuscript is well written and provides much useful information regarding the evolution of ECIG usage and subcultural/geographical differences. Because this investigation presents so much data, the manuscript would greatly benefit from the addition of four more tables (indicated below) comparing groups of participants (i.e. Dutch historical, Dutch current and Belgian current). This would allow the reader to see the information and make comparisons quickly.

è We would like to thank the Reviewer for the careful evaluation and constructive feedback, which have been very helpful in improving our manuscript. In the response to Reviewer, we discuss in detail how we addressed each of the issues that were raised. We hope that you will find this revised version satisfactory. Please let us know if additional changes would be necessary.

Abstract: 

Line 34 states that “the Belgian vapers probably expose themselves to larger quantities of harmful or potentially harmful constituents (HPHCs)”. The word probably is too strong a word and should be replaced with “may have a greater potential to” (or something similar). This would be more in line with what the author’s state later in the manuscript, “until future research indicates a clear link, this hypothesis remains largely speculative” (Line 500).

è We agree with the reviewer and we have made the suggested replacement on line 35.

Methods:

A flowchart showing timelines and procedures used to recruit participants would be helpful.

è We have added a flowchart on line 99.

Results: 

Statistical Analyses and Structure of Presentation of Results (Lines 173 to 188) needs to be included as part of the methods, not the results.

è We moved 3.1. Statistical Analyses and Structure of Presentation of Results to the Method section (now 2.4 Statistical Analyses and Structure of Presentation of Results).

A table showing sociodemographics for Dutch historical (when data is available), Dutch current and Belgian current should be provided for ease of comparison.

A table showing smoking history and status for Dutch historical (when data is available), Dutch current and Belgian current should be provided for ease of comparison.

A table showing vaping parameters (i.e. volume of E-liquid consumed, nicotine used PG/VG ratio used, types of devices used (to include wattage, voltage, temperature, resistance of coils and perhaps a list of the most common flavors preferred)  for Dutch historical (when data is available), Dutch current and Belgian current should be provided for ease of comparison. If available, it would be nice to know the type of coils preferred (i.e. kanthal, nichrome, stainless steel, etc).

A table showing nicotine dependency for Dutch historical (when data is available), Dutch current and Belgian current should be provided for ease of comparison.

The already existing tables should also include data from the Dutch historical group (when data is available)

è As suggested by the Reviewer we have added the four requested tables. The first two (on sociodemographic characteristics and smoking history) were added to the appendix A, since this information was not our main focus. More details on data on these questions for the Dutch Historical group are available in Van Gucht, D.; Adriaens, K.; Baeyens, F. Online vape shop customers who use c-cigarettes report abstinence from smoking and improved quality of life, but a substantial minority still have vaping-related health concerns. Int. J. Environ. Res. Public Health 2017, 14, doi:10.3390/ijerph14070798. We have decided not to add these data to the tables for reasons of readability, parsimony and to avoid duplication. If, however, it is decided that this is necessary, we will adjust this. Tables on vaping parameters (now Table 2) and nicotine dependency (now Table 7) were added to the manuscript. These questions (apart from volume and liquid) were not asked in the Dutch Historical group.

Discussion:

The discussion is long and should be shortened. There is no need to summarize all the results of the present study. Only mention the results required to argue a point (either for or against) with what is already available in the primary literature or to propose a new idea or concept based on the findings of the study.

è We agree with the Reviewer and have shortened our Discussion, by removing the following part from the original manuscript: lines 418-423, 424-427, 429-431, 432-436, 437-438.

Reviewer 3 Report

The manuscript titled “High-Wattage Big-Volume Low-Nicotine Vaping: Subculture or General Trend?” reports the analysis of two online questionnaires carried out among e-cigarette users/smokers who are users of one Belgian and one Dutch website, respectively. The main finding of the study is that while one group uses higher-powered devices with lower nicotine content e-liquid than the other group (lower-powered devices with higher nicotine content), the overall nicotine consumption is at similar levels due to higher amounts of nicotine consumed in the first group. The authors suggest similar findings have been reported previously for Switzerland and the UK and this study adds to the further understanding of the behavior of e-cigarette users, which is needed to inform future regulation of these devices. Further, the manuscript is mostly well-written and I would recommend to publish the manuscript pending several comments below:

1.      While the authors do briefly touch upon this, the authors should consider adding information on the two sources where e-cigarette users were recruited. How similar are the two mentioned websites? E.g., are users particularly interested in high-powered “mod” devices possibly more present in a Facebook sub-group than in an online store? Does the Dutch online store offer e-liquids with “low nicotine content” such as reported by the Belgian Facebook group? Who frequents the Dutch website? Are these new users? There are several such questions on the comparability of the two recruitment pools that should be addressed in the manuscript.

2.      While the authors briefly mention that self-information by users might error-prone, they dismiss it could have an effect on the outcomes of their study (l.541 an following). This goes back to the argument raised above: is it possible that the Facebook group contains mainly e-cigarette users who vape “for fun” and who might be better informed on their devices than the Dutch online shoppers, who might even be using prepared pod systems that often don’t report voltage/wattage settings? Further, previous studies have shown that e-cigarette users often do not know the nicotine content of their e-liquid (or simply assume there is none); how confident are the authors in the reported data?

3.      While the authors report on very different PG/VG ratios used between the two surveyed groups, the manuscript does not contain any discussion on how the ratio of PG/VG might affect the power of the used devices. Generally speaking, VG-heavy e-liquids need a higher power output of the device given its higher boiling point in comparison to PG.

4.      The paragraph on l.424-431 discusses reported health benefits of switching from cigarettes to e-cigarettes. It would be interesting to compare the results of the presented study to studies that report on the health benefits of cigarette users who quit completely (rather than switching to e-cigarettes). Are such reported benefits similar to those reported here?

5.      The paragraph on l.450 discusses the absolute amount of nicotine consumed per week between the two groups, and in comparison to cigarette smokers. The question, however, is the comparability between e-cigarette and cigarette nicotine delivery: does a consumption of ~200mg of nicotine yield the same amount of nicotine delivery in cigarettes and e-cigarettes? And does a high-powered device possibly deliver nicotine more efficiently to the aerosol resulting in different exposures between the two surveyed groups?

6.      L.527 should contain at least a reference, if not an explanation: why have very low nicotine concentrations become the norm, and how does this fit with the increased availability of high-nicotine e-liquids containing “nicotine salt”, such as the popular “Juul” e-cigarette, which has already entered several European markets at high nicotine levels?

Minor comments:

7.      The authors should consider if the chosen title is sufficient to convey the message of their manuscript. It would seem to me that the observed Dutch e-cigarette users did not engage in the described “high-wattage, big-volume, low-nicotine” behavior, nor will the small sample sizes of users recruited on dedicated vaping websites be sufficient to depict a larger population of e-cigarette users.

8.      This is ultimately a question of style, but the authors use of parentheses seems excessive (e.g. lines 70, 73, 78, 529), as well as the use emphasizing words such as very (l.258, l.512).

9.      The word “meaningful” (l. 317) seems odd. Maybe the authors mean “statistically relevant”?

10.   The use of the word “more” (L. 317-330) sounds odd. Maybe authors should consider using “preferential”; e.g. L.322 and following: “…were indicated preferentially by the Belgian…”?

11.   L.523 and following: the word “for” should be replaced in several instances with the word “of”, e.g., fear of nicotine, etc.

12.   The use of the word “decent” should be reviewed (l.84).

13.   The word “smokes” (l.108) likely should read “smokers”

14.   Table 2: The authors should add to the table header what the words “start” and “continue” refer to, as this unclear from looking at Table 2. Also, were all participants asked about these two separate categories, even if they did not vape?

Author Response

The manuscript titled “High-Wattage Big-Volume Low-Nicotine Vaping: Subculture or General Trend?” reports the analysis of two online questionnaires carried out among e-cigarette users/smokers who are users of one Belgian and one Dutch website, respectively. The main finding of the study is that while one group uses higher-powered devices with lower nicotine content e-liquid than the other group (lower-powered devices with higher nicotine content), the overall nicotine consumption is at similar levels due to higher amounts of nicotine consumed in the first group. The authors suggest similar findings have been reported previously for Switzerland and the UK and this study adds to the further understanding of the behavior of e-cigarette users, which is needed to inform future regulation of these devices. Further, the manuscript is mostly well-written and I would recommend to publish the manuscript pending several comments below:

We are grateful for the Reviewer’s positive evaluation and highlighting the usefulness of our findings, we have carefully implemented the specific suggestions made, as detailed below. We hope that you will find this revised version satisfactory. Please let us know if additional changes would be necessary.

1.      While the authors do briefly touch upon this, the authors should consider adding information on the two sources where e-cigarette users were recruited. How similar are the two mentioned websites? E.g., are users particularly interested in high-powered “mod” devices possibly more present in a Facebook sub-group than in an online store? Does the Dutch online store offer e-liquids with “low nicotine content” such as reported by the Belgian Facebook group? Who frequents the Dutch website? Are these new users? There are several such questions on the comparability of the two recruitment pools that should be addressed in the manuscript.

We checked and the web shop (ecig4u) does offer many 0 mg and 3 mg nicotine liquids, apart from several (DIY-mix) Shake&Vapes with recipes for 0-1-2-3 mg/ml. On the other hand, and unlike some Belgian shops, they also have a lot on offer in the 6-12-18 mg/ml nicotine range. Ecig4u now offers one type of prefilled pod devices (with nic salts, 20mg, the Hexa kit https://www.e-cig4u.nl/Hexa-kit), but that is not their main business. They mostly offer “average” wattage devices (10-20-30 watts), but also do have high-wattage vapes (up to 100-200-220 watts). 

We believe that we cannot say much more than we already mention in the manuscript on the web shop customers themselves. Most of them are not “new users” – please see “vaping profile” in the Results section.

2.      While the authors briefly mention that self-information by users might error-prone, they dismiss it could have an effect on the outcomes of their study (l.541 an following). This goes back to the argument raised above: is it possible that the Facebook group contains mainly e-cigarette users who vape “for fun” and who might be better informed on their devices than the Dutch online shoppers, who might even be using prepared pod systems that often don’t report voltage/wattage settings? Further, previous studies have shown that e-cigarette users often do not know the nicotine content of their e-liquid (or simply assume there is none); how confident are the authors in the reported data?

We agree that the Facebook group is a bit “activist” & “heavy metal vaping” oriented, and that they could attract more “high-power-mod” vapers than the web shop. The Belgian Current group might be more “vape-for-fun” people and/or may be better informed. Nevertheless, a lot of this is “guessing” rather than “knowing”.

About the reported nicotine content, we are rather confident (printed on the bottle).

3.      While the authors report on very different PG/VG ratios used between the two surveyed groups, the manuscript does not contain any discussion on how the ratio of PG/VG might affect the power of the used devices. Generally speaking, VG-heavy e-liquids need a higher power output of the device given its higher boiling point in comparison to PG.

Indeed, a higher VG could lead to a choice for selecting a higher wattage because more energy is needed to generate the aerosol with VG than with PG (higher boiling point), but this is probably not the only cause of vaping at high wattage in the Belgian group. For example, the use of low-resistance coils (that usually consist of thicker wire requiring more power to heat up) by the Belgian group also invites a choice for a higher wattage, to achieve a sufficiently high temperature and aerosolization. Moreover, we observed a zero correlation (in the Belgian group) and a (weak) negative correlation (in the Dutch group) between VG percentage and wattage typically used, an observation that goes counter the “high VG results in choice for higher wattage” hypothesis.

4.      The paragraph on l.424-431 discusses reported health benefits of switching from cigarettes to e-cigarettes. It would be interesting to compare the results of the presented study to studies that report on the health benefits of cigarette users who quit completely (rather than switching to e-cigarettes). Are such reported benefits similar to those reported here?

We thank the Reviewer for the suggestion,but since it was not the primary research question in this study, we decided not to go into this in the manuscript. However, if the Reviewer has a particular reference in mind, in which the same or very similar health benefits were questioned, we would gladly incorporate it.

5.      The paragraph on l.450 discusses the absolute amount of nicotine consumed per week between the two groups, and in comparison to cigarette smokers. The question, however, is the comparability between e-cigarette and cigarette nicotine delivery: does a consumption of ~200mg of nicotine yield the same amount of nicotine delivery in cigarettes and e-cigarettes? And does a high-powered device possibly deliver nicotine more efficiently to the aerosol resulting in different exposures between the two surveyed groups?

We thank the Reviewer for this question. We did not focus on speed-of-delivery in our argument, but rather on total “consumption” and how much of that consumption is retained. To clarify this, we have added the following sentence to our revised manuscript starting on line 457 “Research showed that both in vapers [17] and in tobacco smokers [18] the retention rate of nicotine (% of the consumed dose), is around 99% at least when the e-cigarette aerosol or the tobacco smoke is inhaled.”

6.      L.527 should contain at least a reference, if not an explanation: why have very low nicotine concentrations become the norm, and how does this fit with the increased availability of high-nicotine e-liquids containing “nicotine salt”, such as the popular “Juul” e-cigarette, which has already entered several European markets at high nicotine levels?

We thank the Reviewer for the suggestion and added the following paragraph to our revised manuscript, starting on line 528 “Remarkably, and largely in parallel with the "low nicotine high wattage vaping" trend, a diametrically opposite development is currently also taking place in countries outside the TPD - and mainly so in the USA. Namely, since the introduction of the JUUL "pod"-based e-cigarette[30], low-wattage vaping (lower than 10 watts) of liquids with very high concentrations (50 mg/ml or more) of protonated nicotine (containing "nicotine salts") has, starting in 2016, taken a large share of the closed-system e-cigarette market in the USA. TPD forbids the marketing of these devices (and/or high-nicotine liquids) in the European market, even though “down-tuned" pod-based systems and e-liquids with nicotine salts at TPD-compliant levels of 20mg/ml have recently been introduced to the Belgian and Dutch market.” 

Minor comments:

7.      The authors should consider if the chosen title is sufficient to convey the message of their manuscript. It would seem to me that the observed Dutch e-cigarette users did not engage in the described “high-wattage, big-volume, low-nicotine” behavior, nor will the small sample sizes of users recruited on dedicated vaping websites be sufficient to depict a larger population of e-cigarette users.

We have changed the Title into: When Less is More: Vaping Low-Nicotine vs. High-Nicotine E-liquid is Compensated by Increased Wattage and Higher Liquid Consumption

8.      This is ultimately a question of style, but the authors use of parentheses seems excessive (e.g. lines 70, 73, 78, 529), as well as the use emphasizing words such as very (l.258, l.512).

We have removed the parentheses on lines 73, 78, 79, 538 and the emphasizing words on lines 265, 513.

9.      The word “meaningful” (l. 317) seems odd. Maybe the authors mean “statistically relevant”?

We have replaced “meaningful” with “statistically reliable” on line 326.

10.   The use of the word “more” (L. 317-330) sounds odd. Maybe authors should consider using “preferential”; e.g. L.322 and following: “…were indicated preferentially by the Belgian…”?

We have replaced “more” with “more frequently” or with “more often” in this section.

11.   L.523 and following: the word “for” should be replaced in several instances with the word “of”, e.g., fear of nicotine, etc.

We have replaced “for” with “of” on line 524.

12.   The use of the word “decent” should be reviewed (l.84).

We have replaced “decent” with “an extensive and diverse” on line 85.

13.   The word “smokes” (l.108) likely should read “smokers”

We have replaced “smokes” with “smokers” on line 111.

14.   Table 2: The authors should add to the table header what the words “start” and “continue” refer to, as this unclear from looking at Table 2. Also, were all participants asked about these two separate categories, even if they did not vape?

We apologize if the table header was not so clear. We have adapted the title: Table 3. Distribution of responses (in %) regarding the reasons for starting to use e-cigs (all ever vapers) and the reasons for continuing to use e-cigs (all current vapers) for the Dutch Current / Belgian Current group. As mentioned on lines 123-127 “Then, all participants who had ever used an e-cig were asked to indicate the reasons for starting using e-cigs. Current vapers were asked to indicate why they still used e-cigs”. Results showed that (line 244 and following) “Almost all (97%) participants of the Dutch Current group had already tried an e-cig, and 99% of those vaped every day”. This was very comparable in the Belgian Current group.  We have added the correct n’s to the table.